# FINITE-TIME ERROR BOUNDS FOR DISTRIBUTED LINEAR STOCHASTIC APPROXIMATION

## ABSTRACT

This paper considers a novel multi-agent linear stochastic approximation algorithm driven by Markovian noise and general consensus-type interaction, in which each agent evolves according to its local stochastic approximation process which depends on the information from its neighbors. The interconnection structure among the agents is described by a time-varying directed graph. While the convergence of consensus-based stochastic approximation algorithms when the interconnection among the agents is described by doubly stochastic matrices (at least in expectation) has been studied, less is known about the case when the interconnection matrix is simply stochastic. For any uniformly strongly connected graph sequences whose associated interaction matrices are stochastic, the paper derives finite-time bounds on the mean-square error, defined as the deviation of the output of the algorithm from the unique equilibrium point of the associated ordinary differential equation. For the case of interconnection matrices being stochastic, the equilibrium point can be any unspecified convex combination of the local equilibria of all the agents in the absence of communication. Both the cases with constant and time-varying step-sizes are considered. In the case when the convex combination is required to be a straight average and interaction between any pair of neighboring agents may be uni-directional, so that doubly stochastic matrices cannot be implemented in a distributed manner, the paper proposes a push-type distributed stochastic approximation algorithm and provides its finite-time bounds for the performance by leveraging the analysis for the consensus-type algorithm with stochastic matrices.

## 1 INTRODUCTION

The use of reinforcement learning (RL) to obtain policies that describe solutions to a Markov decision process (MDP) in which an autonomous agent interacting with an unknown environment aims to optimize its long term reward is now standard Sutton & Barto (1998). Multi-agent or distributed reinforcement learning is useful when a team of agents interacts with an unknown environment or system and aims to collaboratively accomplish tasks involving distributed decision-making. Distributed here implies that agents exchange information only with their neighbors according to a certain communication graph. Recently, many distributed algorithms for multi-agent RL have been proposed and analyzed (Zhang et al., 2019). The basic result in such works is of the type that if the graph describing the communication among the agents is bi-directional (and hence can be represented by a doubly stochastic matrix), then an algorithm that builds on traditional consensus algorithms converges to a solution in terms of policies to be followed by the agents that optimize the sum of the utility functions of all the agents; further, both finite and infinite time performance of such algorithms can be characterized (Doan et al., 2019; Zhang et al., 2018b).

This paper aims to relax the assumption of requiring bi-directional communication among agents in a distributed RL algorithm. This assumption is arguably restrictive and will be violated due to reasons such as packet drops or delays, differing privacy constraints among the agents, heterogeneous capabilities among the agents in which some agents may be able to communicate more often or with more power than others, adversarial attacks, or even sophisticated resilient consensus algorithms being used to construct the distributed RL algorithm. A uni-directional communication graph can be represented through a (possibly time-varying) stochastic – which may not be doubly stochastic – matrix being used in the algorithm. As we discuss in more detail below, relaxing the assumption of a doubly stochastic matrix to simply a stochastic matrix in the multi-agent and distributed RL algorithms that have been proposed in the literature, however, complicates the proofs of their convergence and finite time performance characterizations. The main result in this paper is to provide a finite time bound on the mean-square error for a multi-agent linear stochastic approximation algorithm in which the agents interact over a time-varying directed graph characterized by a stochastic matrix. This paper, thus, extends the applicability of distributed and multi-agent RL

algorithms presented in the literature to situations such as those mentioned above where bidirectional communication at every time step cannot be guaranteed. As we shall see, this extension is technically challenging and requires new proof techniques that may be of independent interest.

**Related Work** (See Appendix B for more related work) Many distributed reinforcement learning algorithms have now been proposed in the literature. In this setting, each agent can receive information only from its neighbors, and no single agent can solve the problem alone or by 'taking the lead'. A backbone of almost all distributed RL algorithms proposed in the literature is the consensus-type interaction among the agents, dating back at least to Tsitsiklis (1984). Many works have analyzed asymptotic convergence of such RL algorithms using ODE methods (Zhang & Zavlanos, 2019; Zhang et al., 2018b; Suttle et al., 2020; Zhang et al., 2018a). This can be viewed as an application of ideas from distributed stochastic approximation (Kushner & Yin, 1987; Stanković et al., 2010; Huang, 2012; Stanković & Stanković, 2016; Bianchi et al., 2013; Stanković et al., 2016). Finite-time performance guarantees for distributed RL have also been provided in works, most notably in Doan et al. (2019; 2021); Wang et al. (2020); Zhang et al. (2021); Sun et al. (2020); Zeng et al. (2020).

The assumption that is the central concern of this paper and is made in all the existing finite-time analyses for distributed RL algorithms is that the consensus interaction is characterized by doubly stochastic matrices (Doan et al., 2019; 2021; Wang et al., 2020; Zhang et al., 2021; Sun et al., 2020; Zeng et al., 2020) at every time step, or at least in expectation, i.e., $W\mathbf{1} = \mathbf{1}$ and $\mathbf{1}^\top \mathbf{E}(W) = \mathbf{1}^\top$ (Bianchi et al., 2013). Intuitively, doubly stochastic matrices imply symmetry in the communication graph, which almost always requires bidirectional communication graphs. More formally, the assumption of doubly stochastic matrices is restrictive since distributed construction of a doubly stochastic matrix needs to either invoke algorithms such as the Metropolis algorithm (Xiao et al., 2005) which requires bi-directional communication of each agent's degree information; or to utilize an additional distributed algorithm (Gharesifard & Cortés, 2012) which significantly increases the complexity of the whole algorithm design. Doubly stochastic matrices in expectation can be guaranteed via so-called broadcast gossip algorithms which still requires bi-directional communication for convergence (Bianchi et al., 2013). In a realistic network, especially with mobile agents such as autonomous vehicles, drones, or robots, uni-directional communication is inevitable due to various reasons such as asymmetric communication and privacy constraints, non-zero communication failure probability between any two agents at any given time, and application of resilient consensus in the presence of adversary attacks (Vaidya et al., 2012; LeBlanc et al., 2013), all leading to an interaction among the agents characterized by a stochastic matrix, which may further be time-varying. The problem of design of distributed RL algorithms with time-varying stochastic matrices and characterizing either their asymptotic convergence or finite time analysis remains open.

As a step towards solving this problem, we propose a novel distributed stochastic approximation algorithm and provide its convergence analyses when a time-dependent stochastic matrix is being used due to uni-directional communication in a dynamic network. One of the first guarantees to be lost as the assumption of doubly stochastic matrices is removed is that the algorithm converges to a "policy" that maximizes the sum of reward functions of all the agents. Instead, the convergence is to a set of policies that optimize a convex combination of the network-wise accumulative reward, with the exact combination depending on the limit product of the infinite sequence of stochastic matrices. Nonetheless, by defining the error as the deviation of the output of the algorithm from the eventual equilibrium point, we derive finite-time bounds on the mean-square error. We consider both the cases with constant and time-varying step sizes. In the important special case where the goal is to optimize the average of the individual accumulative rewards of all the agents, we provide a distributed stochastic approximation algorithm, which builds on the push-sum idea (Kempe et al., 2003) that has been used to solve distributed averaging problem over strongly connected graphs, and characterize its finite-time performance. Thus, this paper provides the first distributed algorithm that can be applied (e.g., in Temporal difference (TD) learning, see Appendix D) to converge to the policy maximizing the team objective of the sum of the individual utility functions over time-varying, uni-directional, communication graphs, and characterizes the finite-time bounds on the mean-square error of the algorithm output from the equilibrium point under appropriate assumptions.

**Technical Innovation and Contributions** There are two main technical challenges in removing the assumption of doubly stochastic matrices being used in the analysis of distributed stochastic approximation algorithms. The first is in the direction of finite-time analysis. For distributed RL algorithms, finite-time performance analysis essentially boils down to two parts, namely bounding the consensus error and bounding the "single-agent" mean-square error. For the case when consensus

interaction matrices are all doubly stochastic, the consensus error bound can be derived by analyzing the square of the 2-norm of the deviation of the current state of each agent from the average of the states of the agents. With consensus in the presence of doubly stochastic matrices, the average of the states of the agents remains invariant. Thus, it is possible to treat the average value as the state of a fictitious agent to derive the mean-square consensus error bound with respect to the limiting point. More formally, this process relies on two properties of a doubly stochastic matrix $W$, namely that (1) $\mathbf{1}^\top W = \mathbf{1}^\top$, and (2) if $x_{t+1} = W x_t$, then $\|x_{t+1} - (\mathbf{1}^\top x_{t+1})\mathbf{1}\|_2 \leq \sigma_2(W)\|x_t - (\mathbf{1}^\top x_t)\mathbf{1}\|_2$ where $\sigma_2(W)$ denotes the second largest singular value of $W$ (which is strictly less than one if $W$ is irreducible). Even if the doubly stochastic matrix is time-varying (denoted by $W_t$), property (1) still holds and property (2) can be generalized as in Nedić et al. (2018). Thus, the square of the 2-norm $\|x_t - (\mathbf{1}^\top x_t)\mathbf{1}\|_2^2$ is a quadratic Lyapunov function for the average consensus processes. Doubly stochastic matrices in expectation can be treated in the same way by looking at the expectation. This is the core on which all the existing finite-time analyses of distributed RL algorithms are based.

However, if each consensus interaction matrix is stochastic, and not necessarily doubly stochastic, the above two properties may not hold. In fact, it is well known that quadratic Lyapunov functions for general consensus processes $x_{t+1} = S_t x_t$, with $S_t$ being stochastic, do not exist (Olshevsky & Tsitsiklis, 2008). This breaks down all the existing analyses and provides the first technical challenge that we tackle in this paper. Specifically, we appeal to the idea of quadratic comparison functions for general consensus processes. This was first proposed in Touri (2012) and makes use of the concept of "absolute probability sequences". We provide a general analysis methodology and results that subsume the existing finite-time analyses for single-timescale distributed linear stochastic approximation and TD learning as special cases (see Appendix D).

The second technical challenge arises from the fact that with stochastic matrices, the distributed RL algorithms may not converge to the policies that maximize the average of the utility functions of the agents. To regain this property, we propose a new algorithm that utilizes a push-sum protocol for consensus. However, finite-time analysis for such a push-based distributed algorithm is challenging. Almost all, if not all, the existing push-based distributed optimization works build on the analysis in Nedić & Olshevsky (2014); however, that analysis assumes that a convex combination of the entire history of the states of each agent (and not merely the current state of the agent) is being calculated. This assumption no longer holds in our case. To obtain a direct finite-time error bound without this assumption, we propose a new approach to analyze our push-based distributed algorithm by leveraging our consensus-based analyses to establish direct finite-time error bounds for stochastic approximation. Specifically, we tailor an "absolute probability sequence" for the push-based stochastic approximation algorithm and exploit its properties. Such properties have never been found in the existing literature and may be of independent interest for analyzing any push-sum based distributed algorithm.

We now list the main contributions of our work. We propose a novel consensus-based distributed linear stochastic approximation algorithm driven by Markovian noise in which each agent evolves according to its local stochastic approximation process and the information from its neighbors. We assume only a (possibly time-varying) stochastic matrix being used during the consensus phase, which is a more practical assumption when only unidirectional communication is possible among agents. We establish both convergence guarantees and finite-time bounds on the mean-square error, defined as the deviation of the output of the algorithm from the unique equilibrium point of the associated ordinary differential equation. The equilibrium point can be an "uncontrollable" convex combination of the local equilibria of all the agents in the absence of communication. We consider both the cases of constant and time-varying step-sizes. Our results subsume the existing results on convergence and finite-time analysis of distributed RL algorithms that assume doubly stochastic matrices and bi-directional communication as special cases. In the case when the convex combination is required to be a straight average and interaction between any pair of neighboring agents may be uni-directional, we propose a push-type distributed stochastic approximation algorithm and establish its finite-time performance bound. It is worth emphasizing that it is straightforward to extend our algorithm from the straight average point to any pre-specified convex combination. Since it is well known that TD algorithms can be viewed as a special case of linear stochastic approximation (Tsitsiklis & Roy, 1997), our distributed linear stochastic approximation algorithms and their finite-time bounds can be applied to TD algorithms in a straight-forward manner; (see distributed TD($\lambda$) in Appendix D).

**Notation** We use $X_t$ to represent that a variable $X$ is time-dependent and $t \in \{0, 1, 2, \ldots\}$ is the discrete time index. The $i$th entry of a vector $x$ will be denoted by $x^i$ and, also, by $(x)^i$ when convenient. The $ij$th entry of a matrix $A$ will be denoted by $a^{ij}$ and, also, by $(A)^{ij}$ when convenient. We use $\mathbf{1}_n$ to denote the vectors in $\mathbb{R}^n$ whose entries all equal to 1's, and $I$ to denote the identity matrix, whose dimension is to be understood from the context. Given a set $\mathcal{S}$ with finitely many elements, we use $|\mathcal{S}|$ to denote the cardinality of $\mathcal{S}$. We use $\lceil \cdot \rceil$ to denote the ceiling function.

A vector is called a stochastic vector if its entries are nonnegative and sum to one. A square nonnegative matrix is called a row stochastic matrix, or simply stochastic matrix, if its row sums all equal one. Similarly, a square nonnegative matrix is called a column stochastic matrix if its column sums all equal one. A square nonnegative matrix is called a doubly stochastic matrix if its row sums and column sums all equal one. The graph of an $n \times n$ matrix is a direct graph with $n$ vertices and a directed edge from vertex $i$ to vertex $j$ whenever the $ji$-th entry of the matrix is nonzero. A directed graph is strongly connected if it has a directed path from any vertex to any other vertex. For a strongly connected graph $\mathbb{G}$, the distance from vertex $i$ to another vertex $j$ is the length of the shortest directed path from $i$ to $j$; the longest distance among all ordered pairs of distinct vertices $i$ and $j$ in $\mathbb{G}$ is called the diameter of $\mathbb{G}$. The union of two directed graphs, $\mathbb{G}_p$ and $\mathbb{G}_q$, with the same vertex set, written $\mathbb{G}_p \cup \mathbb{G}_q$, is meant the directed graph with the same vertex set and edge set being the union of the edge set of $\mathbb{G}_p$ and $\mathbb{G}_q$. Since this union is a commutative and associative binary operation, the definition extends unambiguously to any finite sequence of directed graphs.

## 2 DISTRIBUTED LINEAR STOCHASTIC APPROXIMATION

Consider a network consisting of $N$ agents. For the purpose of presentation, we label the agents from 1 through $N$. The agents are not aware of such a global labeling, but can differentiate between their neighbors. The neighbor relations among the $N$ agents are characterized by a time-dependent directed graph $\mathbb{G}_t = (\mathcal{V}, \mathcal{E}_t)$ whose vertices correspond to agents and whose directed edges (or arcs) depict neighbor relations, where $\mathcal{V} = \{1, \ldots, N\}$ is the vertex set and $\mathcal{E}_t = \mathcal{V} \times \mathcal{V}$ is the edge set at time $t$. Specifically, agent $j$ is an in-neighbor of agent $i$ at time $t$ if $(j, i) \in \mathcal{E}_t$, and similarly, agent $k$ is an out-neighbor of agent $i$ at time $t$ if $(i, k) \in \mathcal{E}_t$. Each agent can send information to its out-neighbors and receive information from its in-neighbors. Thus, the directions of edges represent the directions of information flow. For convenience, we assume that each agent is always an in- and out-neighbor of itself, which implies that $\mathbb{G}_t$ has self-arcs at all vertices for all time $t$. We use $\mathcal{N}_t^i$ and $\mathcal{N}_t^{i-}$ to denote the in- and out-neighbor set of agent $i$ at time $t$, respectively, i.e.,

$$\mathcal{N}_t^i = \{j \in \mathcal{V} : (j, i) \in \mathcal{E}_t\}, \quad \mathcal{N}_t^{i-} = \{k \in \mathcal{V} : (i, k) \in \mathcal{E}_t\}.$$

It is clear that $\mathcal{N}_t^i$ and $\mathcal{N}_t^{i-}$ are nonempty as they both contain index $i$.

We propose the following distributed linear stochastic approximation over a time-varying neighbor graph sequence $\{\mathbb{G}_t\}$. Each agent $i$ has control over a random vector $\theta_t^i$ which is updated by

$$\theta_{t+1}^i = \sum_{j \in \mathcal{N}_t^i} w_t^{ij} \theta_t^j + \alpha_t \left( A(X_t) \sum_{j \in \mathcal{N}_t^i} w_t^{ij} \theta_t^j + b^i(X_t) \right), \quad i \in \mathcal{V}, \quad t \in \{0, 1, 2, \ldots\}, \quad (1)$$

where $w_t^{ij}$ are consensus weights, $\alpha_t$ is the step-size at time $t$, $A(X_t)$ is a random matrix and $b^i(X_t)$ is a random vector, both generated based on the Markov chain $\{X_t\}$ with state spaces $\mathcal{X}$. It is worth noting that the update of each agent only uses its in-neighbors' information and thus is distributed.

**Remark 1** *The work of Kushner & Yin (1987) considers a different consensus-based networked linear stochastic approximation as follows:*

$$\theta_{t+1}^i = \sum_{j \in \mathcal{N}_t^i} w_t^{ij} \theta_t^j + \alpha_t \left( A(X_t) \theta_t^i + b^i(X_t) \right), \quad i \in \mathcal{V}, \quad t \in \{0, 1, 2, \ldots\}, \quad (2)$$

*whose state form is $\Theta_{t+1} = W_t \Theta_t + \alpha_t \Theta_t A(X_t)^\top + \alpha_t B(X_t)$, and mainly focuses on asymptotically weakly convergence for the fixed step-size case (i.e., $\alpha_t = \alpha$ for all t). Under the similar set of conditions, with its condition (C3.4') being a stochastic analogy for Assumption 6, Theorem 3.1 in Kushner & Yin (1987) shows that equation 2 has a limit which can be verified to be the same as $\theta^*$, the limit of equation 1. How to apply the finite-time analysis tools in this paper to equation 2 has so far eluded us. The two updates equation 1 and equation 2 are analogous to the "combine-then-adapt" and "adapt-then-combine" diffusion strategies in distributed optimization (Chen & Sayed, 2012).* □

We impose the following assumption on the weights $w_t^{ij}$ which has been widely adopted in consensus literature (Jadbabaie et al., 2003; Olfati-Saber et al., 2007; Nedić & Liu, 2017).

**Assumption 1** *There exists a constant $\beta > 0$ such that for all $i, j \in \mathcal{V}$ and $t$, $w_t^{ij} \geq \beta$ whenever $j \in \mathcal{N}_t^i$. For all $i \in \mathcal{V}$ and $t$, $\sum_{j \in \mathcal{N}_t^i} w_t^{ij} = 1$.*

Let $W_t$ be the $N \times N$ matrix whose $ij$th entry equals $w_t^{ij}$ if $j \in \mathcal{N}_t^i$ and zero otherwise. From Assumption 1, each $W_t$ is a stochastic matrix that is compliant with the neighbor graph $\mathbb{G}_t$. Since each agent $i$ is always assumed to be an in-neighbor of itself, all diagonal entries of $W_t$ are positive. Thus, if $\mathbb{G}_t$ is strongly connected, $W_t$ is irreducible and aperiodic. To proceed, define

$$
\Theta_t = \left[ \begin{array}{c} (\theta_t^1)^\top \\ \vdots \\ (\theta_t^N)^\top \end{array} \right], \quad B(X_t) = \left[ \begin{array}{c} (b^1(X_t))^\top \\ \vdots \\ (b^N(X_t))^\top \end{array} \right].
$$

Then, the $N$ linear stochastic recursions in equation 1 can be combined and written as

$$
\Theta_{t+1} = W_t \Theta_t + \alpha_t W_t \Theta_t A(X_t)^\top + \alpha_t B(X_t), \quad t \in \{0, 1, 2, \ldots\}. \tag{3}
$$

The goal of this section is to characterize the finite-time performance of equation 1, or equivalently equation 3, with the following standard assumptions, which were adopted e.g. in Srikant & Ying (2019); Doan et al. (2019).

**Assumption 2** *There exists a matrix $A$ and vectors $b^i$, $i \in \mathcal{V}$, such that*

$$
\lim_{t \to \infty} \mathbf{E}[A(X_t)] = A, \quad \lim_{t \to \infty} \mathbf{E}[b^i(X_t)] = b^i, \quad i \in \mathcal{V}.
$$

*Define $b_{\max} = \max_{i \in \mathcal{V}} \sup_{x \in \mathcal{X}} \|b^i(x)\|_2 < \infty$ and $A_{\max} = \sup_{x \in \mathcal{X}} \|A(x)\|_2 < \infty$. Then, $\|A\|_2 \leq A_{\max}$ and $\|b^i\|_2 \leq b_{\max}$, $i \in \mathcal{V}$.*

**Assumption 3** *Given a positive constant $\alpha$, we use $\tau(\alpha)$ to denote the mixing time of the Markov chain $\{X_t\}$ for which*

$$
\begin{cases} \|\mathbf{E}[A(X_t) - A | X_0 = X]\|_2 \leq \alpha, \quad \forall X, \ \forall t \geq \tau(\alpha), \\ \|\mathbf{E}[b^i(X_t) - b^i | X_0 = X]\|_2 \leq \alpha, \quad \forall X, \ \forall t \geq \tau(\alpha), \ \forall i \in \mathcal{V}. \end{cases}
$$

*The Markov chain $\{X_t\}$ mixes at a geometric rate, i.e., there exists a constant $C$ such that $\tau(\alpha) \leq -C \log \alpha$.*

**Assumption 4** *All eigenvalues of $A$ have strictly negative real parts, i.e., $A$ is a Hurwitz matrix. Then, there exists a symmetric positive definite matrix $P$, such that $A^\top P + P A = -I$. Let $\gamma_{\max}$ and $\gamma_{\min}$ be the maximum and minimum eigenvalues of $P$, respectively.*

**Assumption 5** *The step-size sequence $\{\alpha_t\}$ is positive, non-increasing, and satisfies $\sum_{t=0}^\infty \alpha_t = \infty$ and $\sum_{t=0}^\infty \alpha_t^2 < \infty$.*

To state our first main result, we need the following concepts.

**Definition 1** *A graph sequence $\{\mathbb{G}_t\}$ is uniformly strongly connected if there exists a positive integer $L$ such that for any $t \geq 0$, the union graph $\cup_{k=t}^{t+L-1} \mathbb{G}_k$ is strongly connected. If such an integer exists, we sometimes say that $\{\mathbb{G}_t\}$ is uniformly strongly connected by sub-sequences of length $L$.*

**Remark 2** *Two popular joint connectivity definitions in consensus literature are "B-connected" (Nedić et al., 2009) and "repeatedly jointly strongly connected" (Cao et al., 2008). A graph sequence $\{\mathbb{G}_t\}$ is B-connected if there exists a positive integer $B$ such that the union graph $\cup_{t=kB}^{(k+1)B-1} \mathbb{G}_t$ is strongly connected for each integer $k \geq 0$. Although the uniformly strongly connectedness looks more restrictive compared with B-connectedness at first glance, they are in fact equivalent. To see this, first it is easy to see that if $\{\mathbb{G}_t\}$ is uniformly strongly connected, $\{\mathbb{G}_t\}$ must*

be $B$-connected; now supposing $\{\mathbb{G}_t\}$ is $B$-connected, for any fix $t$, the union graph $\cup_{k=t}^{t+2B-1}\mathbb{G}_k$ must be strongly connected, and thus $\{\mathbb{G}_t\}$ is uniformly strongly connected by sub-sequences of length $2B$. Thus, the two definitions are equivalent. It is also not hard to show that the uniformly strongly connectedness is equivalent to "repeatedly jointly strongly connectedness" provided the graphs under consideration all have self-arcs at all vertices, as "repeatedly jointly strongly connectedness" is defined upon "graph composition". $\qquad\square$

**Definition 2** *Let $\{W_t\}$ be a sequence of stochastic matrices. A sequence of stochastic vectors $\{\pi_t\}$ is an absolute probability sequence for $\{W_t\}$ if $\pi_t^\top = \pi_{t+1}^\top W_t$ for all $t \geq 0$.*

This definition was first introduced by Kolmogorov (Kolmogoroff, 1936). It was shown by Blackwell (Blackwell, 1945) that every sequence of stochastic matrices has an absolute probability sequence. In general, a sequence of stochastic matrices may have more than one absolute probability sequence; when the sequence of stochastic matrices is "ergodic", it has a unique absolute probability sequence (Nedić & Liu, 2017). It is easy to see that when $W_t$ is a fixed irreducible stochastic matrix $W$, $\pi_t$ is simply the normalized left eigenvector of $W$ for eigenvalue one. More can be said.

**Lemma 1** *Suppose that Assumption 1 holds. If $\{\mathbb{G}_t\}$ is uniformly strongly connected, then there exists a unique absolute probability sequence $\{\pi_t\}$ for the matrix sequence $\{W_t\}$ and a constant $\pi_{\min} \in (0, 1)$ such that $\pi_t^i \geq \pi_{\min}$ for all $i$ and $t$.*

Let $\langle\theta\rangle_t = \sum_{i=1}^N \pi_t^i \theta_t^i$, which is a column vector and convex combination of all $\theta_t^i$. It is easy to see that $\langle\theta\rangle_t = (\pi_t^\top \Theta_t)^\top = \Theta_t^\top \pi_t$. From Definition 2 and equation 3, we have $\pi_{t+1}^\top \Theta_{t+1} = \pi_{t+1}^\top W_t \Theta_t + \alpha_t \pi_{t+1}^\top W_t \Theta_t A(X_t)^\top + \alpha_t \pi_{t+1}^\top B(X_t) = \pi_t^\top \Theta_t + \alpha_t \pi_t^\top \Theta_t A(X_t)^\top + \alpha_t \pi_{t+1}^\top B(X_t)$, which implies that

$$\langle\theta\rangle_{t+1} = \langle\theta\rangle_t + \alpha_t A(X_t)\langle\theta\rangle_t + \alpha_t B(X_t)^\top \pi_{t+1}. \tag{4}$$

Asymptotic performance of equation 1 with any uniformly strongly connected neighbor graph sequence is characterized by the following two theorems.

**Theorem 1** *Suppose that Assumptions 1, 2 and 5 hold. Let $\{\theta_t^i\}$, $i \in \mathcal{V}$, be generated by equation 1. If $\{\mathbb{G}_t\}$ is uniformly strongly connected, then $\lim_{t\to\infty} \|\theta_t^i - \langle\theta\rangle_t\|_2 = 0$ for all $i \in \mathcal{V}$.*

Theorem 1 only shows that all the sequences $\{\theta_t^i\}$, $i \in \mathcal{V}$, generated by equation 1 will finally reach a consensus, but not necessarily convergent or bounded. To guarantee the convergence of the sequences, we further need the following assumption, whose validity is discussed in Remark 3.

**Assumption 6** *The absolute probability sequence $\{\pi_t\}$ for the stochastic matrix sequence $\{W_t\}$ has a limit, i.e., there exists a stochastic vector $\pi_\infty$ such that $\lim_{t\to\infty} \pi_t = \pi_\infty$.*

**Theorem 2** *Suppose that Assumptions 1–6 hold. Let $\{\theta_t^i\}$, $i \in \mathcal{V}$, be generated by equation 1 and $\theta^*$ be the unique equilibrium point of the ODE*

$$\dot{\theta} = A\theta + b, \quad b = \sum_{i=1}^N \pi_\infty^i b^i, \tag{5}$$

*where $A$ and $b^i$ are defined in Assumption 2 and $\pi_\infty$ is defined in Assumption 6. If $\{\mathbb{G}_t\}$ is uniformly strongly connected, then all $\theta_t^i$ will converge to $\theta^*$ both with probability 1 and in mean square.*

**Remark 3** *Though Assumption 6 may look restrictive at first glance, simple simulations show that the sequences $\{\theta_t^i\}$, $i \in \mathcal{V}$, do not converge if the assumption does not hold. It is worth emphasizing that the existence of $\pi_\infty$ does not imply the existence of $\lim_{t\to\infty} W_t$, though the converse is true. Indeed, the assumption subsumes various cases including (a) all $W_t$ are doubly stochastic matrices, and (b) all $W_t$ share the same left eigenvector for eigenvalue 1, which may arise from the scenario when the number of in-neighbors of each agent does not change over time (Olshevsky & Tsitsiklis, 2013). An important implication of Assumption 6 is when the consensus interaction among the agents, characterized by $\{W_t\}$, is replaced by resilient consensus algorithms such as Vaidya et al. (2012); LeBlanc et al. (2013) in order to attenuate the effect of unknown malicious*

*agents, the resulting dynamics of non-malicious agents, in general, will not converge, because the resulting interaction stochastic matrices among the non-malicious agents depend on the state values transmitted by the malicious agents, which can be arbitrary, and thus the resulting stochastic matrix sequence, in general, does not have a convergent absolute probability sequence; of course, in this case, the trajectories of all the non-malicious agents will still reach a consensus as long as the step-size is diminishing, as implied by Theorem 1. Further discussion on Assumption 6 can be found in Appendix C.* □

We now study the finite-time performance of the proposed distributed linear stochastic approximation equation 1 for both fixed and time-varying step-size cases. Its finite-time performance is characterized by the following theorem.

Let $\eta_t = \|\pi_t - \pi_\infty\|_2$ for all $t \geq 0$. From Assumption 6, $\eta_t$ converges to zero as $t \to \infty$.

**Theorem 3** *Let the sequences $\{\theta_t^i\}$, $i \in \mathcal{V}$, be generated by equation 1. Suppose that Assumptions 1–4, 6 hold and $\{\mathbb{G}_t\}$ is uniformly strongly connected by sub-sequences of length $L$. Let $q_t$ and $m_t$ be the unique integer quotient and remainder of $t$ divided by $L$, respectively. Let $\delta_t$ be the diameter of $\cup_{k=t}^{t+L-1} \mathbb{G}_k$, $\delta_{\max} = \max_{t \geq 0} \delta_t$, and*

$$\epsilon = \left(1 + \frac{2b_{\max}}{A_{\max}} - \frac{\pi_{\min}\beta^{2L}}{2\delta_{\max}}\right)(1 + \alpha A_{\max})^{2L} - \frac{2b_{\max}}{A_{\max}}(1 + \alpha A_{\max})^L, \qquad (6)$$

*where $0 < \alpha < \min\{K_1, \frac{\log 2}{A_{\max}\tau(\alpha)}, \frac{0.1}{K_2\gamma_{\max}}\}$.*

**1) Fixed step-size:** *Let $\alpha_t = \alpha$ for all $t \geq 0$. For all $t \geq T_1$,*

$$\sum_{i=1}^N \pi_t^i \mathbf{E}\left[\left\|\theta_t^i - \theta^*\right\|_2^2\right] \leq 2\epsilon^{q_t} \sum_{i=1}^N \pi_{m_t}^i \mathbf{E}\left[\left\|\theta_{m_t}^i - \langle\theta\rangle_{m_t}\right\|_2^2\right] + C_1\left(1 - \frac{0.9\alpha}{\gamma_{\max}}\right)^{t-T_1} + C_2$$

$$+ \frac{\gamma_{\max}}{\gamma_{\min}} 2\alpha\zeta_4 \sum_{k=0}^{t-T_1} \eta_{t+1-k}\left(1 - \frac{0.9\alpha}{\gamma_{\max}}\right)^k. \qquad (7)$$

**2) Time-varying step-size:** *Let $\alpha_t = \frac{\alpha_0}{t+1}$ with $\alpha_0 \geq \frac{\gamma_{\max}}{0.9}$. For all $t \geq LT_2$,*

$$\sum_{i=1}^N \pi_t^i \mathbf{E}\left[\left\|\theta_t^i - \theta^*\right\|_2^2\right] \leq 2\epsilon^{q_t-T_2} \sum_{i=1}^N \pi_{LT_2+m_t}^i \mathbf{E}\left[\left\|\theta_{LT_2+m_t}^i - \langle\theta\rangle_{LT_2+m_t}\right\|_2^2\right]$$

$$+ C_3\left(\alpha_0\epsilon^{\frac{q_t-1}{2}} + \alpha_{\lceil\frac{q_t-1}{2}\rceil L}\right) + \frac{1}{t}\left(C_4 \log^2\left(\frac{t}{\alpha_0}\right) + C_5 \sum_{k=LT_2}^t \eta_k + C_6\right). \qquad (8)$$

*Here $T_1, T_2, K_1, K_2, C_1 - C_6$ are finite constants whose definitions are given in Appendix A.1.*

Since $\pi_t^i$ is uniformly bounded below by $\pi_{\min} \in (0, 1)$ from Lemma 1, it is easy to see that the above bound holds for each individual $\mathbf{E}[\|\theta_t^i - \theta^*\|_2^2]$. To better understand the theorem, we provide the following remark.

**Remark 4** *In Appendix E.2.1, we show that both $\epsilon$ and $(1 - \frac{0.9\alpha}{\gamma_{\max}})$ lie in the interval $(0, 1)$. It is easy to show that $\epsilon$ is monotonically increasing for $\delta_{\max}$ and $L$, monotonically decreasing for $\beta$ and $\pi_{\min}$. Also, $\lim_{t\to\infty} \sum_{k=0}^{t-T_1} \eta_{t+1-k}(1 - \frac{0.9\alpha}{\gamma_{\max}})^k \leq \lim_{t\to\infty} \frac{\gamma_{\max}}{0.9\alpha}[\eta_{\lceil\frac{t-T_1}{2}\rceil} + \eta_1(1 - \frac{0.9\alpha}{\gamma_{\max}})^{\frac{t-T_1}{2}}] = 0$. Therefore, the summands in the finite-time bound equation 7 for the fixed step-size case are exponentially* decaying except for *the constant $C_2$, which implies that $\limsup_{t\to\infty} \sum_{i=1}^N \pi_t^i \mathbf{E}[\|\theta_t^i - \theta^*\|_2^2] \leq C_2$, providing a constant limiting bound. From Appendix A, $C_2$ is monotonically increasing for $\gamma_{\max}, \delta_{\max}, b_{\max}$ and $L$, and monotonically decreasing for $\gamma_{\min}, \pi_{\min}$ and $\beta$. In Appendix E.2.2, we show that $\lim_{t\to\infty} \frac{1}{t} \sum_{k=1}^t \eta_k = 0$, which implies that the finite-time bound equation 8 for the time-varying step-size case converges to zero as $t \to \infty$. We next comment on $0.1$ in the inequality defining $\alpha$. Actually, we can replace $0.1$ with any constant $c \in (0, 1)$, which will affect the value of $\epsilon$ and the feasible set of $\alpha$, with the latter becoming $0 < \alpha < \min\{K_1, \frac{\log 2}{A_{\max}\tau(\alpha)}, \frac{c}{K_2\gamma_{\max}}\}$. Thus,*

*the smaller the value of $c$ is, the smaller is the feasible set of $\alpha$, though the feasible set is always nonempty. For convenience, we simply pick $c = 0.1$ in this paper; that is why we also have $0.9$ in equation 7. Lastly, we comment on $\alpha_0$ in the time-varying step-size case. We set $\alpha_0 \geq \frac{\gamma_{\max}}{0.9}$ for the purpose of getting a cleaner expression of the finite-time bound. For $\alpha_0 < \frac{\gamma_{\max}}{0.9}$, our approach still works, but will yield a more complicated expression. The same is true for Theorem 5.* □

**Technical Challenge and Proof Sketch**  As described in the introduction, the key challenge of analyzing the finite-time performance of the distributed stochastic approximation equation 1 lies in the condition that the consensus-based interaction matrix is time-varying and stochastic (not necessarily doubly stochastic). To tackle this, we appeal to the absolute probability sequence $\pi_t$ of the time-varying interaction matrix sequence and introduce the quadratic Lyapunov comparison function $\sum_{i=1}^{N} \pi_t^i \mathbf{E}[\|\theta_t^i - \theta^*\|_2^2]$. Then, using the inequality $\sum_{i=1}^{N} \pi_t^i \mathbf{E}[\|\theta_t^i - \theta^*\|_2^2] \leq 2 \sum_{i=1}^{N} \pi_t^i \mathbf{E}[\|\theta_t^i - \langle\theta\rangle_t\|_2^2] + 2\mathbf{E}[\|\langle\theta\rangle_t - \theta^*\|_2^2]$, the next step is to find the finite-time bounds of $\sum_{i=1}^{N} \pi_t^i \mathbf{E}[\|\theta_t^i - \langle\theta\rangle_t\|_2^2]$ and $\mathbf{E}[\|\langle\theta\rangle_t - \theta^*\|_2^2]$, respectively. The latter term is essentially the "single-agent" mean-square error. Our main analysis contribution here is to bound the former term for both fixed and time-varying step-size cases.

## 3    PUSH-SA

The preceding section shows that the limiting state of consensus-based distributed stochastic approximation depends on $\pi_\infty$, which leads to a convex combination of the local equilibria of all the agents in the absence of communication, but the convex combination is in general "uncontrollable". Note that this convex combination will correspond to a convex combination of the network-wise accumulative rewards in applications such as distributed TD learning. In an important case when the convex combination is desired to be the straight average, the existing literature e.g. Doan et al. (2019; 2021) relies on doubly stochastic matrices whose corresponding $\pi_\infty = (1/N)\mathbf{1}_N$. As mentioned in the introduction, doubly stochastic matrices implicitly require bi-directional communication between any pair of neighboring agents; see e.g. gossiping (Boyd et al., 2006) and the Metropolis algorithm (Xiao et al., 2005). A popular method to achieve the straight average target while allowing uni-directional communication between neighboring agents is to appeal to the idea so-called "push-sum" (Kempe et al., 2003), which was tailored for solving the distributed averaging problem over directed graphs and has been applied to distributed optimization (Nedić & Olshevsky, 2014). In this section, we will propose a push-based distributed stochastic approximation algorithm tailored for uni-directional communication and establish its finite-time error bound.

Each agent $i$ has control over three variables, namely $y_t^i$, $\tilde{\theta}_t^i$ and $\theta_t^i$, in which $y_t^i$ is scalar-valued with initial value 1, $\tilde{\theta}_t^i$ can be arbitrarily initialized, and $\theta_0^i = \tilde{\theta}_0^i$. At each time $t \geq 0$, each agent $i$ sends its weighted current values $\hat{w}_t^{ji} y_t^i$ and $\hat{w}_t^{ji}(\tilde{\theta}_t^i + \alpha_t A(X_t)\theta_t + \alpha_t b^i(X_t))$ to each of its current out-neighbors $j \in \mathcal{N}_t^{i-}$, and updates its variables as follows:

$$\begin{cases} y_{t+1}^i = \sum_{j \in \mathcal{N}_t^i} \hat{w}_t^{ij} y_t^j, \qquad y_0^i = 1, \\[2ex] \tilde{\theta}_{t+1}^i = \sum_{j \in \mathcal{N}_t^i} \hat{w}_t^{ij} \left[ \tilde{\theta}_t^j + \alpha_t \left( A(X_t)\theta_t^j + b^j(X_t) \right) \right], \\[2ex] \theta_{t+1}^i = \dfrac{\tilde{\theta}_{t+1}^i}{y_{t+1}^i}, \qquad \theta_0^i = \tilde{\theta}_0^i, \end{cases} \qquad (9)$$

where $\hat{w}_t^{ij} = 1/|\mathcal{N}_t^{j-}|$. It is worth noting that the algorithm is distributed yet requires that each agent be aware of the number of its out-neighbors.

Asymptotic performance of equation 9 with any uniformly strongly connected neighbor graph sequence is characterized by the following theorem.

**Theorem 4**  *Suppose that Assumptions 2–5 hold. Let $\{\theta_t^i\}$, $i \in \mathcal{V}$, be generated by equation 9 and $\theta^*$ be the unique equilibrium point of the ODE*

$$\dot{\theta} = A\theta + \frac{1}{N} \sum_{i=1}^{N} b^i, \qquad (10)$$

where $A$ and $b^i$ are defined in Assumption 2. If $\{\mathbb{G}_t\}$ is uniformly strongly connected, then $\theta_t^i$ will converge to $\theta^*$ in mean square for all $i \in \mathcal{V}$.

In this section, we define $\langle\tilde{\theta}\rangle_t = \frac{1}{N}\sum_{i=1}^N \tilde{\theta}_t^i$ and $\langle\theta\rangle_t = \frac{1}{N}\sum_{i=1}^N \theta_t^i$. To help understand these definitions, let $\hat{W}_t$ be the $N \times N$ matrix whose $ij$-th entry equals $\hat{w}_t^{ij}$ if $j \in \mathcal{N}_t^i$, otherwise equals zero. It is easy to see that each $\hat{W}_t$ is a column stochastic matrix whose diagonal entries are all positive. Then, $\pi_t = \frac{1}{N}\mathbf{1}_N$ for all $t \geq 0$ can be regarded as an absolute probability sequence of $\{\hat{W}_t\}$. Thus, the above two definitions are intuitively consistent with $\langle\theta\rangle_t$ in the previous section.

Finite-time performance of equation 9 with any uniformly strongly connected neighbor graph sequence is characterized by the following theorem.

Let $\mu_t = \|A(X_t)(\langle\theta\rangle_t - \langle\tilde{\theta}\rangle_t)\|_2$. In Appendix E.3, we show that $\|\langle\theta\rangle_t - \langle\tilde{\theta}\rangle_t\|_2$ converges to zero as $t \to \infty$, so does $\mu_t$.

**Theorem 5** *Suppose that Assumptions 2–4 hold and $\{\mathbb{G}_t\}$ is uniformly strongly connected by subsequences of length $L$. Let $\{\theta_t^i\}$, $i \in \mathcal{V}$, be generated by equation 9 with $\alpha_t = \frac{\alpha_0}{t+1}$ and $\alpha_0 \geq \frac{\gamma_{\max}}{0.9}$. Then, there exists a nonnegative $\bar{\epsilon} \leq (1 - \frac{1}{N^{NL}})^{\frac{1}{L}}$ such that for all $t \geq \bar{T}$,*

$$\sum_{i=1}^N \mathbf{E}\left[\left\|\theta_{t+1}^i - \theta^*\right\|_2^2\right] \leq C_7\bar{\epsilon}^t + C_8\left(\alpha_0\bar{\epsilon}^{\frac{t}{2}} + \alpha_{\lceil\frac{t}{2}\rceil}\right) + C_9\alpha_t$$

$$+ \frac{1}{t}\left(C_{10}\log^2\left(\frac{t}{\alpha_0}\right) + C_{11}\sum_{k=\bar{T}}^t \mu_k + C_{12}\right), \tag{11}$$

*where $\bar{T}$ and $C_7 - C_{12}$ are finite constants whose definitions are given in Appendix A.2.*

In Appendix E.3, we show that $\lim_{t\to\infty}\frac{1}{t}\sum_{k=1}^t \mu_k = 0$, which implies that the finite-time bound equation 11 converges to zero as $t \to \infty$. It is worth mentioning that the theorem does not consider the fixed step-size case, as our current analysis approach cannot be directly applied for this case.

**Proof Sketch and Technical Challenge** Using the inequality $\sum_{i=1}^N \mathbf{E}[\|\theta_{t+1}^i - \theta^*\|_2^2] \leq 2\sum_{i=1}^N \mathbf{E}[\|\theta_{t+1}^i - \langle\tilde{\theta}\rangle_t\|_2^2] + 2N\mathbf{E}[\|\langle\tilde{\theta}\rangle_t - \theta^*\|_2^2]$, our goal is to derive the finite-time bounds of $\sum_{i=1}^N \mathbf{E}[\|\theta_{t+1}^i - \langle\tilde{\theta}\rangle_t\|_2^2]$ and $\mathbf{E}[\|\langle\tilde{\theta}\rangle_t - \theta^*\|_2^2]$, respectively. Although this looks similar to the proof of Theorem 3, the derivation is quite different. First, the iteration of $\langle\tilde{\theta}\rangle_t$ is a single-agent SA plus a disturbance term $\langle\theta\rangle_t - \langle\tilde{\theta}\rangle_t$, so we cannot directly apply the existing single-agent SA finite-time analyses to bound $\mathbf{E}[\|\langle\tilde{\theta}\rangle_t - \theta^*\|_2^2]$; instead, we have to show that $\langle\theta\rangle_t - \langle\tilde{\theta}\rangle_t$ will diminish and quantify the diminishing "speed". Second, both the proof of showing diminishing $\langle\theta\rangle_t - \langle\tilde{\theta}\rangle_t$ and derivation of bounding $\sum_{i=1}^N \mathbf{E}[\|\theta_{t+1}^i - \langle\tilde{\theta}\rangle_t\|_2^2]$ involve a key challenge: to prove the sequence $\{\theta_t^i\}$ generated from the Push-SA equation 9 is bounded almost surely. To tackle this, we introduce a novel way to constructing an absolute probability sequence for the Push-SA as follows. From equation 9, $\theta_{t+1}^i = \sum_{j=1}^N \tilde{w}_t^{ij}[\theta_t^j + \alpha_t A(X_t)\frac{\theta_t^j}{y_t^j} + \alpha_t\frac{b^j(X_t)}{y_t^j}]$, where $\tilde{w}_t^{ij} = (\hat{w}_t^{ij}y_t^j)/(\sum_{k=1}^N \hat{w}_t^{ik}y_t^k)$. We show that each matrix $\tilde{W}_t = [\tilde{w}_t^{ij}]$ is stochastic, and there exists a unique absolute probability sequence $\{\tilde{\pi}_t\}$ for the matrix sequence $\{\tilde{W}_t\}$ such that $\tilde{\pi}_t^i \geq \tilde{\pi}_{\min}$ for all $i \in \mathcal{V}$ and $t \geq 0$, with the constant $\tilde{\pi}_{\min} \in (0, 1)$. Most importantly, we show two critical properties of $\{\tilde{W}_t\}$ and $\{\tilde{\pi}_t\}$, namely $\lim_{t\to\infty}(\Pi_{s=0}^t \tilde{W}_s) = \frac{1}{N}\mathbf{1}_N\mathbf{1}_N^\top$ and $\frac{\tilde{\pi}_t^i}{y_t^i} = \frac{1}{N}$ for all $i, j \in \mathcal{V}$ and $t \geq 0$, which have never been reported in the literature though push-sum based algorithms have been extensively studied.

## 4 CONCLUDING REMARKS

In this paper, we have established both asymptotic and non-asymptotic analyses for a consensus-based distributed linear stochastic approximation algorithm over uniformly strongly connected graphs, and proposed a push-based variant for coping with uni-directional communication. Both algorithms and their analyses can be directly applied to TD learning. One limitation of our finite-time bounds is that they involve quite a few constants which are well defined and characterized but whose values are not easy to compute. Future directions include leveraging the analyses for resilience in the presence of malicious agents and extending the tools to more complicated RL.

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
