# OpenReview forum: "Finite-Time Error Bounds for Distributed Linear Stochastic Approximation"
_ICLR.cc/2022/Conference — ICLR 2022 Submitted_

### Official Review · Reviewer_HPzY · 2021-10-29

**Correctness:** 3
**Technical Novelty And Significance:** 4
**Empirical Novelty And Significance:** 4
**Recommendation:** 5
**Confidence:** 5

**Main Review:**

Originality: This paper studies distributed Markovian stochastic approximation algorithms with stochastic interconnection matrices, which has not yet been covered in the related literature. The results are a bit different than what is known for related algorithms having doubly stochastic interconnection matrices. Further, a push-type algorithm is proposed for decentralized implementation.

Quality. The results are technically sound and theoretically supported. The algorithm proposed is fresh and new as well as interesting.

Clarity. The paper is well written and easy to follow. The results are nicely organized and the contributions relative to existing works are clear too.

Significance. The results are important and complement the existing distributed Markovian stochastic approximation results which have only focused on doubly stochastic interconnections among agents. The tools developed for analysis may also be useful for further use.

**Summary Of The Paper:**

An interesting submission dealing with distributed stochastic approximation driven by Markovian noise, in which the communication topologies considered among agents are captured by a stochastic matrix (in contrast to the doubly stochastic matrix studied in the related literature). Both asymptotic and finite-time error bounds are established and it was shown that the algorithm converges to some unspecified convex combination of the equilibrium points of local tasks. This paper finally proposes a push-type distributed stochastic approximation algorithm and provides its finite-time bounds for the performance by leveraging the analysis for the consensus-type algorithm with stochastic matrices. All analysis and error bounds are directly applicable to distributed TD algorithms.



**Summary Of The Review:**

Numerical simulations are suggested to compare the proposed algorithm and existing ones as well as validate the proved finite-time error bounds for both constant/time-varying stepsizes.

--------------
After rebuttal: After reading the other reviews and authors response, I have updated my score.

---

> ### Author Response · Authors · 2021-11-23
> **Response to Reviewer HPzY**
>
> Thanks for the very positive comments.
> We have added a set of simulations by applying our theoretical results to distributed TD learning and comparing with the existing results in the appendix (Section Appendix D).

---

### Official Review · Reviewer_5MZ3 · 2021-11-01

**Correctness:** 3
**Technical Novelty And Significance:** 2
**Empirical Novelty And Significance:** 2
**Recommendation:** 5
**Confidence:** 4

**Main Review:**

The manuscript considers the linear stochastic approximation problem (Srikant & Ying (2019)) for a multi-agent system. The authors revisit similar setups and results as Doan et al. (2019) for the cases where the agents interact over time-varying directed communication networks. Below, the strong and weak points of the paper are pointed out.

Strengths:
- The authors propose a push-sum counterpart for the distributed linear stochastic approximation under time-varying directed networks.
- The manuscript made a nontrivial contribution to generalize the convergence result in Doan et al. (2019) for a more complicated communication setup, time-varying directed networks.
- The authors provide a detailed analysis in the appendix for both proposed algorithms.

Weaknesses:
- The lack of any theoretical comparisons to the literature prevents the readers from understanding the technical differences between the results of this work and the existing theoretical results (e.g., Doan et al. (2019)). The reviewer suggests providing a detailed table, comparing the results in this work in terms of the constants shown in the convergence rate. This is a crucial part of such a theoretical paper.
- The authors introduced RL and specifically TD as examples of the theory developed in this paper. This reviewer suggests adding an illustrative synthetic example over a sequence of directed graphs besides additional discussions on the two proposed algorithms for that example.
- Another serious issue with this work is Assumption 6. Even after reading Appendix C, this reviewer is not convinced that Assumption 6 is a proper assumption. Indeed, the limit in Assumption 6 does not exist for multiple real-world problems. For example, consider a periodic change of $\pi_t$, given periodic communication networks. Given that all agents reach consensus on the same $\theta_t^i$, this reviewer suggests dropping Assumption 6 and modifying the statement in Theorem 2 to some time-varying ODE with $\pi_t^i$ in the definition.
- This reviewer finds the introduction lengthy, yet not sufficiently informative. Most of the discussions address the issues for time-varying directed networks, while the discussions on the linear stochastic approximation are limited. This reviewer strongly encourages the authors to add examples and/or motivations on the importance of this problem for TD (or generally RL) problems.
- As a minor issue, before using an acronym, please first mention the full form, for example, TD learning which is mentioned several times, is the acronym for temporal difference learning.
- One of the main issues of this manuscript is that it takes the reader a long time to find the main objective, which is reaching consensus on the equilibrium point of some ODE, and why this is important. The authors elaborate on the setup details, and after proposing the first algorithm (Eq. (1)), they mention that local $\theta_t^i$ converges to $\theta^\star$ which is the solution to some ODE problem (Eq. (5)). The authors do not explain why reaching consensus on the solution of the ODE is the main objective, i.e., it is not quite clear where the ODE appears in this problem. To understand the role of the ODE, the reader is required to read the Appendix.
- For a non-expert reader, finding the differences between the two proposed algorithms (Eq.(1) and Eq, (9) is not easy. This reviewer suggests the authors take some time to improve the presentation of the two algorithms and highlight the advantages of one to the other (e.g., in a table). The manuscript in its current format is confusing, in the sense that the importance of the second algorithm compared to the first algorithm is vague. This reviewer believes that the authors should specifically address the cases wherein a column stochastic matrix is not guaranteed. This requires clear explanations.
- On a more technical note, it is not clear to this reviewer why in Eq. (1) the update rule is established based on $b^i$, while in Eq.(9) considers a convex combination of neighbors’ $b^j$.

**Summary Of The Paper:**

This paper studies the problem of distributed linear stochastic approximation for a group of agents over time-varying directed communication networks (to be more specific, the setup is decentralized). The authors propose two decentralized algorithms, (1) consensus-based linear stochastic approximation using row-stochastic mixing matrices (Eq. (1)), and (2) a push-sum type algorithm using column-stochastic mixing matrices (Eq. (9)). They further provide results on the asymptotic and finite-time mean-square errors to the equilibrium point of the (proper) ODE for each proposed algorithm.

**Summary Of The Review:**

In summary, this work extends the existing results on the finite-time error bounds for the problem of distributed (decentralized) linear stochastic approximation to a more challenging communication setup. The paper is marginally novel and the contributions are fair. However, the manuscript has two main issues, (1) the comparisons to the previous works are missing, and (ii) the structure of the paper, as well as the presentation of its results, require some work.

---

> ### Author Response · Authors · 2021-11-23
> **Response to Reviewer 5MZ3 (Part 1 of 2)**
>
> (1) We had clearly stated in the introduction (see Technical Innovation and Contributions on page 2) our theoretical contributions compared with the existing literature, including Doan et al. (2019). Specifically, the existing literature, including Doan et al. (2019), only considered the doubly stochastic matrix case, which cannot be applied in the more general cases in our paper. We propose two algorithms. The first is the consensus-based one equation 1 with time-varying stochastic matrices; it subsumes Doan et al. (2019) as a special case when the linear stochastic approximation is TD learning and all stochastic matrices are doubly stochastic. The second is the push-based algorithm equation 9 which can deal with directed graphs with uni-directional communications, whereas all existing algorithms including Doan et al. (2019) cannot.
>
> Since these differences are easy to explain in wording, we do not provide a table comparison. It is worth emphasizing that our contribution is to investigate more general distributed settings, not to "beat" the existing convergence rates. For the special case when considering TD learning and all interaction matrices are doubly stochastic, our results in Theorem 3 simplifies to the results in Doan et al. (2021).
>
> (2) We have added a simulation in Appendix D to show that the distributed TD($\lambda$) learning (a special case of linear stochastic approximation) does not converge even with a periodic communication network.
>
> When Assumption does not hold in some real-world problems, our simulations (see Appendix D.4) show that in those cases, the distributed algorithm (1) and its applications (see TD learning in Appendix D) will not converge to a fixed point. This is a big difference compared with distributed optimization processes. We thus propose the push-base algorithm which does not suffer this limitation and can work for any time-varying directed graphs with the mild assumption (i.e., uniformly strongly connected).
>
>
> Time-varying ODE is a very interesting idea when Assumption 6 does not hold. We actually have thought about it for a while but do not have any concrete idea to analyze and characterize such a non-convergent process.
>
> Convergence to limit sets of time-varying ODEs is described in Chapter 8.2.6. of "Stochastic Approximation and Recursive Algorithms and Applications" by Kushner and Yin.
> Intuitively, the rate of change of the average stochastic process must be very slow for convergence to take place.
> Here this rate of change is "uncontrollable" as it is determined by the network change. If we want to apply the existing theory on time-varying ODEs, we need to additionally assume that the underlying graph changes in a very slow manner, which could also be "restrictive and unrealistic". In our setting, though Assumption 6 looks restrictive, we allow the graph to change at every time step.
>
> (3) The "long" detailed explanation is important (and probably necessary) to motivate this paper. We found that many readers will think extending from doubly stochastic cases to general stochastic and push-based cases is straightforward using the existing tools from distributed optimization. But it turns out that it is not the case. Distributed stochastic approximation brings additional challenges (see details in our response to Reviewer dShT17). Thus, such explanation helps readers to understand the background, motivation, and contributions of the paper, especially for those not familiar with distributed algorithm analysis.
>
> We have added a new section in the appendix (Section Appendix D Distributed TD Learning) to show that distributed stochastic approximation can be directly applied to distributed TD learning.
> We also have added a set of simulations by applying our theoretical results to distributed TD learning and comparing them with the existing results in the appendix (Section Appendix D).
>
> (4) Reaching a consensus at $\theta^*$ among multiple agents in a network means that all agents can learn the optimal point via collaboration. Note that $\theta^*$ depends on all agents' information, so none can learn it only by itself. The necessity of such collaboration motivates the study of distributed algorithms over networks including distributed RL and optimization. We have added in Appendix D a concrete application of our distributed algorithms, namely distributed TD learning. It is shown that all agents can collaboratively perform the optimal evaluation of the whole network policy, where $\theta^*$ represents the desired optimal point, which is the unique stable equilibrium of the ODE (equation 5). In the classic asymptotic analysis of stochastic approximation processes, ODEs are used to describe the limiting behaviors of the processes; see e.g.Borkar (2008).

---

> > ### Author Response · Authors · 2021-11-23
> > **Response to Reviewer 5MZ3 (Part 2 of 2)**
> >
> > (5) The consensus-based algorithm (1) is a generalization of the existing ones, which extends doubly stochastic matrices to more general (time-varying) stochastic matrices. From Theorem 2, its convergence point depends on a convex combination of all agents information $b^i$, while the convex combination weights are not "controllable" as they are determined by the underlying graphs.
> > In order to "control" the convex combination weights in general time-varying directed graphs (where doubly stochastic matrices are not feasible), we propose a push-based algorithm (9). From Theorem 4, the limiting convex combination is always a straight average. Actually, as claimed in Appendix C.3, this limiting convex combination can be any pre-specified one; this generalization is straightforward, so we omitted it in the paper.
> >
> > (6) According to the description of the push-based algorithm (see the paragraph above equation 9, each agent $i$ sends its weighted current values $\hat w_t^{ji} y^i_{t}$ and $\hat w_t^{ji}(\tilde\theta^i_{t} + \alpha_t A(X_t) \theta_t + \alpha_t b^i(X_t)) $ to each of its current out-neighbors. Thus, when $j$ is an in-neighbor of agent $i$, agent $i$ will receive $\hat w_t^{ij}(\tilde\theta^j_{t} + \alpha_t A(X_t) \theta_t + \alpha_t b^j(X_t)) $ from agent $j$, and use this received value in its update (equation 9). This is why $b^j$ appears in the equation 9, but agent $i$ does not need to know $b^j$.

---

### Official Review · Reviewer_fcYJ · 2021-11-02

**Correctness:** 3
**Technical Novelty And Significance:** 3
**Empirical Novelty And Significance:** 1
**Recommendation:** 5
**Confidence:** 3

**Main Review:**

Strength:
               Attacking a hard setting.

Weakness:
1. A major weakness of the paper is the lack of numerical examples in the distributed TD setting. Furthermore, even a simpler toy example demonstrating the interplay between the various quantities in the Theorems is missing.

2. As the paper mentions, the value of many constants cannot be computed, however, it is sort of the inherent difficulty of the setting itself. While it is not a weakness that should limit the evaluation of the paper per se, it is nevertheless important to be highlighted and I thank the authors for the same.



Possible Technical Issues: It will be great if the authors can clarify the following.

* (Question 1) Why is that $b^i$ keeps changing with agents and yet $A$ is the same across agents? To elaborate, let us consider TD with linear function approximation, wherein $\phi(s)$ is the feature for state $s$, then $A=\phi(s_t)\left(\beta\phi(s_{t+1})-\phi(s_t)\right)^\top$, which is based on the transition from state $s_t$ to state $s_{t+1}$. In a distributed RL like setting, different agents could possibly be transitioning from different states, i.e., we would have $s^i_t$ to state $s^i_{t+1}$ and hence $A^i=\phi(s^i_t)\left(\beta\phi(s^i_{t+1})-\phi(s^i_t)\right)^\top$. This is where it would have helped to have a motivating example.



*  (Question 2)  Consider the stochastic approximation in $2$-dimensions for just a single agent. Let $A=\left[\begin{matrix}-\gamma_{\max} ,0 ; 0 ,-\gamma_{\min}\end{matrix}\right]$ and $b=\left[\begin{matrix}0 ;0  \end{matrix}\right]$. In this case $\theta_*=\left[\begin{matrix}0 \\ 0 \end{matrix}\right]$, and assume $\theta_0=\left[\begin{matrix}1 \\ 1  \end{matrix}\right]$. In below, step sizes $\alpha_t=\frac{\alpha_0}{t+1}$, and $\theta_t=\left[\begin{matrix}\theta_t(1) \\ \theta_t(2)  \end{matrix}\right]\in \mathbb{R}^2$.

$\theta_{t+1}(1)=\theta_t(1)-\alpha_t\gamma_{\max}\theta_t(1),$

$\theta_{t+1}(2)=\theta_t(2)-\alpha_t\gamma_{\min}\theta_t(2),$

Let us just look at $\theta_{t+1}(2)$,

$\theta_{t+1}(2)=\Pi_{k=0}^{t}\left(1-\frac{\gamma_{\min}\alpha_0}{k+1}\right) \theta_{0}(2)$

$|\theta_{t+1}(2)|=\Pi_{k=0}^{t}\left(1-\frac{\gamma_{\min}\alpha_0}{k+1}\right)|\theta_{0}(2)|=\Pi_{k=0}^{t}(1-\frac{\gamma_{\min}\alpha_0}{k+1})$

Now using the fact that for for small $x$, we have $e^{-2x}\leq(1-x)$ we have (for sufficiently small $\gamma_{\min}\rightarrow 0$ and for $t\geq 1$):

$|\theta_{t+1}(2)|\geq e^{-2\gamma_{\min}\alpha_0\sum_{k=0}^{t}\frac{1}{k+1} }$

$=e^{-2\gamma_{\min}\alpha_0}e^{-2\gamma_{\min}\alpha_0\sum_{k=1}^{t}\frac{1}{k+1} }$

$\geq e^{-2\gamma_{\min}\alpha_0}e^{-2\gamma_{\min}\alpha_0\ln(t) }$

$= \frac{e^{-2\gamma_{\min}\alpha_0}}{t^{2\gamma_{\min}\alpha_0} }$

Picking $\alpha_0= \frac{\gamma_{\max}}{0.9}$, we have


$|\theta_{t+1}(2)| \geq \frac{e^{-2\gamma_{\min}\alpha_0}}{t^{\frac{2\gamma_{\min}\gamma_{\max}}{0.9}} }$

We can always let $\gamma_{\max}=1$, to get $\frac{1}{t^{\frac{2\gamma_{\min}}{0.9}}}$, which is arbitrarily small rate for arbitrarily small $\gamma_{\min}$. However, Theorem 3-(2) seems to suggest that we are indeed achieving a $\frac1t$ rate. Please clarify the same.


*  (Question 3) We can think of the constant step size case to be $\alpha_t=\alpha_0=\alpha$. Now, it is intriguing that for the constant step size case we have $\alpha_0<\min\\{K_1,\frac{\log 2}{A_{\max}\tau(\alpha)}, \frac{0.1}{K_2\gamma_{\max}}\\}$, where $\gamma_{\max}$ appears in the denominator and for the time varying case we have $\alpha_0\geq \frac{\gamma_{\max}}{0.9}$, where $\gamma_{\max}$ appears in the numerator. Why is this difference on the dependence of step size on $\gamma_{\max}$?



**Summary Of The Paper:**

 The paper presents a finite time analysis of distributed linear stochastic approximation (non-doubly stochastic interconnection matrix) in the presence of Markovian Noise.

**Summary Of The Review:**

While the possible technical issues can be discusses further, the score is mainly due to lack of empirical section which is a major weakness of the paper.

---

> ### Author Response · Authors · 2021-11-23
> **Response to Reviewer fcYJ**
>
> (1) To our knowledge, all the existing distributed stochastic approximation and TD learning works assume all agents share the same $A$ matrix, e.g., Kushner \& Yin (1987) and Doan et al. (2019; 2021).
>
> Even for more complicated RL settings, recent distributed RL development also assumes the network state is shared among all agents, e.g., Zhang et al. (2018b), even though each agent has its own local state.
> From a real-world application perspective, possible applications of cooperative RL include (1) Starcraft’s multi-player mode, (2) UAV fleet control, and (3) financial trading. For (1), players can directly observe state information of their allies, yet reward information such as resources, technologies, and bonuses are generally hidden. For (2), imagine a UAV fleet composed of multiple drones, each of which has complete observability (via radar, GPS, etc.) of the area of operations, but which keep their typically complex reward/utility information private. In (3), consider a group of traders within a single investment firm cooperating to maximize overall return, where all traders observe the same market information and other traders' orders, yet keep their precise gains and losses secret. Note that in all these situations state information is globally observable, while rewards are purely local.
>
> (2) Reviwer fcYJ probably misunderstood the definition of $\gamma_{\max}$ and $\gamma_{\min}$. As defined in Assumption 4, $\gamma_{\max}$ and $\gamma_{\min}$ are the largest and smallest eigenvalues of $P$, not $A$.
>
> Along with the reviewer's idea and example, let
> $A = [- a_{\max}, 0 ; 0, - a_{\min}]$, then $ P = [\frac{1}{2 a_{\max}}, 0 ; 0, \frac{1}{2 a_{\min}}] $ with $\gamma_{\max} = \frac{1}{2 a_{\min}} $ and $\gamma_{\min} = \frac{1}{2 a_{\max}} $.
> From the reviewer's derivation, we have
> $|\theta_{t+1}(2)| \ge  \frac{e^{-2 a_{\min} \alpha_0}}{t^{2a_{\min}\gamma_{\max}/0.9 }} = \frac{e^{-1/0.9}}{t^{1/0.9 }}$. From equation 8, the convergence rate is $\frac{(\log(t))^2}{t}$. Now we show that $t^{-1/0.9}$ converges to $0$ faster than that of $\frac{(\log(t))^2}{t}$. To see this, for any $k<1$,
> \begin{align*}
>     \lim_{t\to\infty} \frac{t^{-k}}{\frac{(\log(t))^2}{t}} =
>     \lim_{t\to\infty} \frac{t^{1-k}}{(\log(t))^2}=
>     \lim_{t\to\infty} \frac{(1-k)t^{1-k}}{2 \log(t)} = \lim_{t\to\infty} \frac{(1-k)^2 t^{1-k}}{2} = 0.
> \end{align*}
> Thus, this is not a counterexample of our theorem.
>
> (3) Note that $\alpha$ is the fixed stepsize and $\alpha_0$ is the time-varying stepsize at initial time 0. From the statement of Theorem 3, the upper bound $\min\{K_1, \frac{\log 2}{A_{\max \tau(\alpha)}}, \frac{0.1}{K_2 \gamma_{\max} }\}$ is only for $\alpha$, not for $\alpha_0$. Thus, $\alpha$ and $\alpha_0$ have independent dependence on $\gamma_{{\rm max}}$.
>
> In general, stepsizes, no matter fixed or time-varying ones, need to be small enough so that the process will convergence. That is why we have an upper bound for the fixed stepsize $\alpha$ in Theorem 3. For time-varying stepsizes, they are assumed to be decreasing, we only need to find an upper bound for the initial stepsize $\alpha_0$. This upper bound is given via Theorem 3 (see the definition of $T_2$ given in Appendix A.1). As for the lower bound requirement $\alpha_0 > \frac{\gamma_{\max}}{0.9}$, it is because the initial stepsize cannot be too small; otherwise the process will not converge either as the time-varying stepsize is diminishing.

---

> > ### Comment · Reviewer_fcYJ · 2021-11-24
> > **Competitive or Collaborative? and Strong Assumption**
> >
> > Thanks for the clarifications. It will be great if the following are also elaborated.
> >
> > (1) If the agents are collaborating, and okay with sharing parameters (via consensus), why have the rewards private or why does it hurt to share the rewards? In other words, it understandable having the rewards strictly private (but states common) in a competitive setting, but in a collaborative one.
> >
> > (2) By letting $\gamma_{\max}=\frac{1}{a_{\min}}$, we are assuming that we know the minimum eigenvalue of $A$ (at least in case when there are no imaginary components in the eigenvalues). Is this not a very strong assumption?

---

> > > ### Author Response · Authors · 2021-11-24
> > > **Response to Reviewer fcYJ**
> > >
> > > (1) Even in a collaborative setting, agents do not want to directly share their rewards perhaps due to more immediate, lower-level privacy concerns, which occur in scenarios like social networks and multi-player financial trading. For example, rewards in a social network may contain individuals' private information like incomings. As another example, a group of traders within a single investment firm cooperate to maximize overall return, while all traders observe other traders' orders, yet keep their precise gains and losses secret. In addition, sharing rewards may harm the privacy of an agent's reward function, which is the goal of inverse reinforcement learning. Of course, how to effectively counter a sophisticated, dedicated effort to reconstruct an agent's reward function in our setting could be a hard problem. It is worth noting that many recent works in collaborative multi-agent reinforcement learning impose the same private local reward assumption, e.g., ICML papers Zhang et al. (2018) and Doan et al. (2019).
> > >
> > > (2) $A$ is assumed to be a stable matrix (Assumption 4), which is the only assumption on $A$. In this case, the Lyapunov equation $A^\top P+PA=-I$ has a unique positive definite solution $P$. Thus, once $A$ is fixed, so are  $\gamma_{{\rm max}}$ and $\gamma_{{\rm min}}$. We do not need to know the value of $A$ and its spectrum, as $\gamma_{{\rm max}}$ and $\gamma_{{\rm min}}$ are finite constants, which are uniquely determined by $A$, and thus do not influence the order of non-asymptotic (finite-time) convergence rate.
> > >
> > > In your special diagonal $A$ matrix, we can derive the explicit expressions of $\gamma_{{\rm max}}$ and $\gamma_{{\rm min}}$ in terms of diagonal entries of $A$. For general stable $A$, it may not be possible, as $P=\int_{0}^\infty e^{A^\top t}e^{At}dt$.

---

> > > > ### Comment · Reviewer_fcYJ · 2021-11-25
> > > > **Minimum Eigenvalue**
> > > >
> > > > Agreed that $A$ needs to be stable is the only (explicit) assumption. However, to choose $\alpha_0=\frac{\gamma_{\max}}{0.9}$, we require $\gamma_{\max}$, and as the authors point out, for a general stable $A$ it may not be possible to have an explicit expression for $\gamma_{\max}$, which implies that $\gamma_{\max}$ has to be computed first and then plugged into $\alpha_0=\frac{\gamma_{\max}}{0.9}$ to achieve the desired convergence rates. How do we compute $\gamma_{\max}$? Is it faster than rates of the main SA itself? It would be great if the authors can comment on this.

---

> > > > > ### Author Response · Authors · 2021-11-25
> > > > > **Response to Reviewer fcYJ**
> > > > >
> > > > > It is a very good question about the inequality $\alpha_0\ge\frac{\gamma_{\max}}{0.9}$. The inequality is not a necessary condition; we impose it in order to simplify the expression of the finite-time bound in equation 8.
> > > > > In the case when the inequality does not hold,
> > > > > we also can derive a finite-time bound using a similar analysis techniques; however, the expression of the finite-time bound will be much more complicated. (Note that algorithm (1) always asymptotically converge for any positive $\alpha_0$; see Theorem 2.)
> > > > > We chose $\alpha_0\ge\frac{\gamma_{\max}}{0.9}$ for simplicity.
> > > > >
> > > > > From the preceding discussion, picking an $\alpha_0$ does not necessarily need to compute $\gamma_{\max}$ in implementation. Even if it is needed, for a general SA, $P$ can be computed by solving the Lyapunov equation ($P=\int_{0}^\infty e^{A^\top t}e^{At}dt$), and then $\gamma_{\max}$ can be obtained by computing $P$'s spectrum. Existing computing software, e.g. MATLAB, has very efficient and fast algorithms to solve Lyapunov equations and compute eigenvalues. These computations apparently need the value of matrix $A$. It is worth emphasizing that when we apply SA to TD learning, the corresponding expression of $A$ is given in terms of known parameters; see Appendix D.1 equation 24. Moreover, in TD($\lambda$), as mentioned in Appendix D.1, $A$ in equation 24 must be a negative definite matrix by Theorem 1 in Tsitsiklis and Roy (1997); thus, one can always choose $P=I$ to satisfy a Lyapunov equation,
> > > > > which means that we can set $\gamma_{\max} = \gamma_{\min} = 1$ no matter what the value of $A$ is in TD($\lambda$). The same holds for TD(0).

---

### Official Review · Reviewer_RnJ8 · 2021-11-04

**Correctness:** 3
**Technical Novelty And Significance:** 3
**Empirical Novelty And Significance:** Not applicable
**Recommendation:** 5
**Confidence:** 4

**Main Review:**

The topic of the proposed paper is certainly significant, and the extension to the general non-doubly stochastic case is important. The proofs appear to be solid at first glance, although I have certainly not verified all 47 pages of the supplementary material. Despite the significance of the topic, however, the paper is not well-conceived or well-written.

For example, the introduction is 3 pages long and is mostly talking about how doubly stochastic matrices are not good enough to represent certain systems and discussing related literature. While it is appropriate to place the results in the context of existing literature, much of this exposition is repetitive and is undercut by the fact that the authors fail to provide a numerical example/simulation illustrating a non-doubly stochastic model. Furthermore, this extensive introduction means the authors have less space to define the problem and present the results.

This brings up the second problem with the paper, which is the lack of exposition and rigour in the definition of the problem and presentation of results. Specifically, the problem being solved is not defined. The exposition begins on page 4 with the proposed approximation/consensus rules. Yet it is unclear what the purpose of these dynamics is and what problem they are solving. I had to do significant literature survey to even understand what problem the authors were trying to solve. This lack of exposition continues to the main results in Thms 3/5. These results are presented as inequalities proving convergence of the proposed dynamics in a certain sense. Yet it is unclear to me exactly in what sense these inequalities prove convergence or what that convergence would imply about whether the proposed dynamics solve a particular problem. For example, the approximation state $\theta$ appears on both the left and the right. Does the inequality imply that the approximation state is converging to the true state or not.

This brings up my third concern which is that a certain lack of rigour impedes understanding, verification and interpretation of the results. For example, the theorem statements are not self-contained and rely on implicit assumptions and definitions of variables mentioned earlier in the exposition. This is particularly problematic since several variables are defined differently at different times. For example, $A$ is initially part of the definition of the dynamics, but then is later taken to be some property of the sequence $\pi_t$. As a result, there are numerous terms in the Thm statements for which must guess at the meaning. This is further impeded by a length series of assumptions included in the Thm statement which depend on terms which do not actually appear in the theorem statement. See the notes for specific concerns.

My final concern is that there is no validation or illustration of the proposed approach on a numerical example. This is rather unusual and makes it particularly difficult for the reader to verify or interpret the results.

- Is assumption 1 required to hold for any $X_t$?
- What are all these assumptions on? The use of assumptions is generally not rigorous, but the particular usage here does not even make it clear what these properties of which variables are being assumed.
- the $\forall X$ in Assumption 3 is unclear.
- Which $A$ is used in assumption 3.
- Which $A$ is used in assumption 4. Also assumption 4 has some discussion which is not part of the assumption.
- The statement of Thm1 is not complete
- The use of both fixed and time-varying step sizes are unclear. I was initially under the impression that the time-varying step could be unknown or stochastic, but it seems to be prescribed. What is the motivation here?
- What is the diameter of a graph in Thm 3.
- why does $i$ not appear in the definition of $\hat w_{t}^{ij}$
- Typos: "decaying expect for" "decreasing" "that consensus interaction" "cannot be directly apply"



**Summary Of The Paper:**

This paper addresses a consensus problem in stochastic approximation, which has application to policy iteration in reinforcement learning. Specifically, the authors address the case where the interaction graph between agents is stochastic, but not doubly stochastic. Under certain assumptions, the authors prove convergence of their proposed algorithm in a certain sense.

**Summary Of The Review:**

In summary, the paper may contain publishable results, but is not sufficiently clear to understand what those results are, let alone verify them. In addition, the lack of a numerical illustration seriously undercuts the claims of significance. To be publishable, the authors should focus on clearly defining the problem, presenting the solution and its significance, and illustrating on a numerical example.

---

> ### Author Response · Authors · 2021-11-23
> **Response to Reviewer RnJ8**
>
> (1) We treat this work as pure theoretical work and thus did not provide any simulations. It is worth noting that quite a few important recent finite-time analysis papers (e.g., Srikant \& Ying (2019) in COLT, Doan et al. (2019) in ICML, Wu et al. (2020) in NeurIPS) do not contain any simulations either.
>
> The "long" detailed explanation is important (and probably necessary) to motivate this paper. We found that many readers will think extending from doubly stochastic cases to general stochastic and push-based cases is straightforward using the existing tools from distributed optimization. But it turns out that it is not the case. Distributed stochastic approximation brings additional challenges (see details in our response to Reviewer dShT17). Thus, such explanation helps readers to understand the background, motivation, and contributions of the paper, especially for those not familiar with distributed algorithm analysis.
>
> Our initial submission contains complete problem formulations and theoretical results. All the constants in the theorems are now postponed to an appendix section (Section Appendix A), which was suggested by one of our earlier submission reviewers. In the revised manuscripts, we have added a set of simulations by applying our theoretical results to distributed TD learning and comparing them with the existing results. We still put them in the appendix (Section Appendix D) as they are not the main contributions of this theoretical works. Also, note that the TD learning section is pretty long. We are flexible if the reviewers feel that the new TD learning section should (partially) replace some introduction parts in the paper. All materials including appendices will be published together anyway, if accepted.
>
> (2) As explained in the introduction (see Page 1), the goal of this paper is to relax the doubly stochastic matrix assumption in the existing distributed stochastic approximation (including TD learning) algorithms and thus expand their applicability in more realistic networks. To this end, we proposed two novel algorithms, one is consensus-based and the other is push-based. Once the algorithms are given, the problem is to provide their asymptotic and finite-time analyses.
>
> As our two main contributions, Theorems 3 and 5 respectively provide the finite-time performance of the consensus-based algorithm (1) and push-based algorithm (9). One important implication of Theorem 3 and its inequality (8) is that algorithm (1) will converge to the unique equilibrium in the mean square because the right-hand side of inequality (8) converges to zero as time $t$ goes to infinity. This is stated in Remark 4 and Theorem 2. A similar implication can be drawn from Theorem 5 and its inequality (11), which was also stated after Theorem 5 and in Theorem 4.
>
> (3) Assumption 1 is imposed for weights $w^{ij}_t$ in the algorithm (1), so it holds whenever the paper talks about the algorithm (1). Please note that for all our theorems, we specify which assumptions are needed.
>
> We believe the initial submission contains all necessary discussions on each assumption and related variables. We also specify which assumptions are needed in every theorem.
>
> $X$ is the initial state of the Markov chain. Why it is unclear?
>
> There is only one $A$ matrix is defined in the paper, which is given in Assumption 2.
>
> The discussion in Assumption 4 is for the purpose of introducing two important parameters $\gamma_{\max}$ and $\gamma_{\min}$. This way makes them easier to refer to.
>
> Fixed and time-varying stepsizes are two basic settings in stochastic approximation. They are not unknown or stochastic, and usually need to be tuned in implementation. That is why they need to satisfy certain conditions, e.g, Assumption 5 and those specifications in Theorems 3, 5, 6 and 7.
>
> The definition of the diameter of a directed graph is given in Notation on page 4.
>
> It is because agent $j$ evenly splits its values among its out-neighbors. This is probably the simplest way to set weights. The weights can be dependent on $i$, as long as their summation equals one and have a uniformly positive lower bound.
>
> We have corrected the typos, except for "decreasing". Could you please specify why it is a typo?
>
> Can you clarify what do you mean by "The statement of Thm1 is not complete"?

---

> > ### Comment · Reviewer_RnJ8 · 2021-11-27
> > **still unclear**
> >
> > The issues raised in my review remain unresolved. I find the paper lacks what I would consider to be mathematical rigour. I pointed out several specific examples, along with more extensive explanations -- but none of these have been addressed. If it were a matter of one or two points, and the authors had made some effort towards addressing those points, I would be willing to try and explain in more detail, but since no effort has been made, I am not convinced further effort on my part would be fruitful.

---

> > > ### Author Response · Authors · 2021-11-27
> > > **Response to Reviewer RnJ8**
> > >
> > > We don't understand what makes you feel "none of these have been addressed", especially about "mathematical rigour". In case you missed our earlier response to your "mathematical rigour" related comments. We now respond again in the following point-to-point manner, which is the same content as part (3) in our last response.
> > >
> > > Q1: $A$ is initially part of the definition of the dynamics, but then is later taken to be some property of the sequence $\pi_t$.
> > > Which $A$ is used in assumption 3?
> > > Which $A$ is used in assumption 4? Also, assumption 4 has some discussion which is not part of the assumption.
> > >
> > > A1: There is only one $A$ matrix is defined in the paper, which is given in Assumption 2.
> > > The discussion in Assumption 4 is for the purpose of introducing two important parameters $\gamma_{\max}$ and $\gamma_{\min}$. This way makes them easier to refer to.
> > >
> > > Q2: Is assumption 1 required to hold for any?
> > >
> > > A2: Assumption 1 is imposed for weights $w^{ij}_t$ in the algorithm (1), so it holds whenever the paper talks about the algorithm (1). Please note that for all our theorems, we specify which assumptions are needed.
> > >
> > > Q3: What are all these assumptions on? The use of assumptions is generally not rigorous, but the particular usage here does not even make it clear what these properties of which variables are being assumed.
> > >
> > > A3: We believe the initial submission contains all necessary discussions on each assumption and related variables. We also specify which assumptions are needed in every theorem.
> > >
> > > Q4: The $\forall X$ in Assumption 3 is unclear.
> > >
> > > A4: $X$ is the initial state of the Markov chain. Why it is unclear?
> > >
> > > Q5: The use of both fixed and time-varying step sizes are unclear. I was initially under the impression that the time-varying step could be unknown or stochastic, but it seems to be prescribed. What is the motivation here?
> > >
> > > A5: Fixed and time-varying stepsizes are two basic settings in stochastic approximation. They are not unknown or stochastic, and usually need to be tuned in implementation. That is why they need to satisfy certain conditions, e.g, Assumption 5 and those specifications in Theorems 3, 5, 6, and 7.
> > >
> > > Q6: What is the diameter of a graph in Thm 3.
> > >
> > > A6: The definition of the diameter of a directed graph is given in Notation on page 4.
> > >
> > > Q7: why does $i$ not appear in the definition of $\hat \omega_t^{ij}$
> > >
> > > A7: It is because agent $j$ evenly splits its values among its out-neighbors. This is probably the simplest way to set weights. The weights can be dependent on $i$, as long as their summation equals one and have a uniformly positive lower bound.
> > >
> > > Q8: Typos: "decaying expect for" "decreasing" "that consensus interaction" "cannot be directly apply"
> > >
> > > A8: We have corrected the typos, except for "decreasing". Could you please specify why it is a typo?
> > >
> > > Q9: The statement of Thm1 is not complete.
> > >
> > > A9: Can you clarify what do you mean by "The statement of Thm1 is not complete"?

---

> > > > ### Comment · Reviewer_RnJ8 · 2021-11-27
> > > > **…..**
> > > >
> > > > Please highlight any changes made to the paper to address the concerns raised above.

---

> > > > > ### Author Response · Authors · 2021-11-27
> > > > > **Response to Reviewer RnJ8**
> > > > >
> > > > > In the revised manuscript, we have highlighted the changes in blue for the typos. We found the remaining concerns are easy to address as they had been clearly stated in the initial submission (see our responses), and thus feel unnecessary to make changes. It seems that the reviewer did not carefully read our paper and responses. Please note that in our responses, we have spotted which parts of the paper the reviewer should refer to answer the reviewer's questions. Please also note that we cannot understand a couple of the reviewer's questions which need clarification.

---

### Official Review · Reviewer_dShT · 2021-11-08

**Correctness:** 4
**Technical Novelty And Significance:** 3
**Empirical Novelty And Significance:** 3
**Recommendation:** 5
**Confidence:** 3

**Main Review:**

The paper studies a concrete problem in distributed optimization in a multi-agent setting, where consensus-type dynamics prevail to allow various agents reach a global equilibrium point, which is often the globally optimal solution of some convex problem. Maintaining a consensus algorithm using bi-directional communication is rather straightforward and, by now, well-understood, but as the authors argue, relaxing this assumption poses some challenge.

The paper is well-written and well-organized. Problem formulation and results are presented clearly and precisely. Though generally well-organized, the paper uses 3.5 (out of 9) pages of the main text for introductory, non-technical part, which suggests that the organization is more suitable for a journal submission than a conference. In this limited review time, I was unable to check the proofs. Nonetheless, the derived error bounds make perfect sense, and they look correct.

My main concern about the paper is that it is marginally relevant for a learning conference like ICLR. To establish connection to ML and RL, the authors provide pointers to some literature studying distributed multi-agent RL problems using distributed optimization frameworks. First, I personally believe that this literature does not represent well distributed, multi-agent RL. Second, the paper addresses a challenge existing in generic distributed optimization, but not in RL. In other words, this challenge does not arise because of the RL nature of the problem, but rather comes from the networked nature of the generic distributed optimization. The problem studied here is a good fit for networked/distributed optimization venues. As such, it might get a reasonable attention if published in NeurIPS and ICML, but I think it will receive much less attention from the ICLR community. This is the main reason behind my low score.


Minor points:

- terms such as “mean-square” and “mean squared” are used to denote the same notion.

- p. 3: double stochastic matrix => doubly stochastic matrix


**Summary Of The Paper:**

This paper studies multi-agent distributed optimization under general consensus-type interactions between agents. The main contribution of the paper is to study linear stochastic approximation in a multi-agent setup without using bi-directional communication among agents. Non-asymptotic upper bounds on the associated error function, in a mean squared sense, are derived.

**Summary Of The Review:**

This is a well-written paper studying a multi-agent distributed optimization under general consensus-type interactions between agents. I personally think that it is not of relevance for ICLR, while it might be on interest to some larger learning conferences such as NeurIPS.

---

> ### Author Response · Authors · 2021-11-23
> **Response to Reviewer dShT**
>
> We have added a new section in the appendix (Section Appendix D Distributed TD Learning) to show that distributed stochastic approximation can be directly applied to distributed TD learning, which is an important policy evaluation problem in RL.
>
> Although the expressions of our distributed linear stochastic approximation algorithms share similarities with distributed optimization, there are fundamental differences and challenges here due to stochastic approximation. First, for the consensus-based algorithm, we need additional Assumption 6, which is not needed for distributed optimization. In distributed optimization, the only requirement for the sequence of time-varying stochastic matrices is their backward product converges to a rank-one matrix. But for distributed stochastic approximation, it is not enough, and additionally we need its corresponding absolute probability sequence $\pi_t$ has a limit (i.e., Assumption 6); otherwise, the convergence of the network cannot be guaranteed (see Figure 1). Second, for the push-based algorithm, its convergence requires an additional boundedness step (Lemma 19), which is not needed for distribution optimization problems. Because of this, we found that existing analysis tools for push-based optimization algorithms cannot solve this boundedness issue, and proposed a novel analysis tool for the push-based algorithm using absolute probability sequences and their unrevealed properties (Lemma 15). All these challenges come from stochastic approximation processes.
>
> We believe that the paper is a good fit for ICLR. Finite-time analysis works were published in previous ICLR, e.g., "Greedy-GQ with Variance Reduction: Finite-time Analysis and Improved Complexity." by Ma, Chen, Zhou and Zou.  It may also be suitable for NeurIPS and ICML. The paper was rejected by NeurIPS because of a debatable criticism on Assumption 6. That is why we have an appendix section (Section Appendix C) to discuss the assumption necessity. We submitted to ICLR as some finite-time analysis papers were published here (e.g. "Greedy-GQ with Variance Reduction: Finite-time Analysis and Improved Complexity." by Ma, Chen, Zhou, and Zou). If the paper is unfortunately rejected by ICLR, any suggestions on venues other than NeurIPS and ICML will be very appreciated. We treat this work as a purely theoretical work and notice that earlier important works in this line do not contain any simulations and were published in various venues including NeurIPS (Wu et al., 2020), ICML (Doan et al., 2019), and COLT (Srikant \& Ying, 2019).

---

### Author Response · Authors · 2021-11-23
**Response to all the reviewers and AC**

Thanks to all the reviewers for their helpful suggestions and comments. We have addressed all their comments and concerns, and highlighted all changes in blue in the revised manuscripts. The main changes include:

1. A new section has been added in the appendix (Section Appendix D Distributed TD Learning) which applies our proposed consensus- and push-based stochastic approximation algorithms and finite-time analyses to TD learning, a fundamental and important policy evaluation problem in RL. We focus on TD($\lambda$) and TD(0) can be done in almost the same way. Note that the existing distributed TD(0) finite-time analysis in Doan et al. (2019) only considers i.i.d. samples. Since TD learning is a special case of linear stochastic approximation, theoretical finite-time bound results are applied immediately. We then validate our theorems by simulations and compare our results with the existing distributed TD($\lambda$) algorithms.

2. We have added a simulation to demonstrate that Assumption 6 is not restrictive.

---

### Decision · Program_Chairs · 2022-01-20

**Decision:**

Reject

**Comment:**

This paper studies a stochastic approximation framework for multi-agent consensus algorithms driven by Markovian noise in the spirit of the classical paper of Kushner & Yin. The authors' main result is that - modulo a series of assumptions, some conceptual, some technical - the generated sequence of play reaches a consensus, and they also estimate the rate of this convergence.

Even though the paper's premise is interesting, the reviewers identified several weaknesses in the paper, and the reviewers that raised them where not convinced by the authors' replies (especially regarding the relative lack of numerical evidence to demonstrate the claims that are not supported by the theory, such as the role of Assumption 6). After my own reading of the paper and the discussion with the reviewers during the rebuttal phase, I concur that this version of the paper does not clear the bar for acceptance - but, at the same time, I would encourage the authors to submit a suitably revised version at the next opportunity.